# Infarct in the Heart: What’s MMP-9 Got to Do with It?

**DOI:** 10.3390/biom11040491

**Published:** 2021-03-25

**Authors:** Mediha Becirovic-Agic, Upendra Chalise, Michael J. Daseke, Shelby Konfrst, Jeffrey D. Salomon, Paras K. Mishra, Merry L. Lindsey

**Affiliations:** 1Center for Heart and Vascular Research, Department of Cellular and Integrative Physiology, University of Nebraska Medical Center, Omaha, NE 68102, USA; mediha.agic@unmc.edu (M.B.-A.); upendra.chalise@unmc.edu (U.C.); john.daseke@unmc.edu (M.J.D.II); shelby.konfrst@unmc.edu (S.K.); jeffrey.salomon@unmc.edu (J.D.S.); paraskumar.mishra@unmc.edu (P.K.M.); 2Research Service, Nebraska-Western Iowa Health Care System, Omaha, NE 68198, USA; 3Department of Physiology and Biophysics, University of Mississippi Medical Center, Jackson, MS 39216, USA; 4Division of Pediatric Critical Care, Department of Pediatrics, University of Nebraska Medical Center, Omaha, NE 68198, USA

**Keywords:** inflammation, extracellular matrix, matrix metalloproteinases, remodeling, macrophage, neutrophil

## Abstract

Over the past three decades, numerous studies have shown a strong connection between matrix metalloproteinase 9 (MMP-9) levels and myocardial infarction (MI) mortality and left ventricle remodeling and dysfunction. Despite this fact, clinical trials using MMP-9 inhibitors have been disappointing. This review focuses on the roles of MMP-9 in MI wound healing. Infiltrating leukocytes, cardiomyocytes, fibroblasts, and endothelial cells secrete MMP-9 during all phases of cardiac repair. MMP-9 both exacerbates the inflammatory response and aids in inflammation resolution by stimulating the pro-inflammatory to reparative cell transition. In addition, MMP-9 has a dual effect on neovascularization and prevents an overly stiff scar. Here, we review the complex role of MMP-9 in cardiac wound healing, and highlight the importance of targeting MMP-9 only for its detrimental actions. Therefore, delineating signaling pathways downstream of MMP-9 is critical.

## 1. Introduction

Myocardial infarction (MI) occurs with prolonged ischemia due to coronary artery occlusion, resulting in irreversible cell death of cardiomyocytes [1,2]. Approximately every 40 s, an American will have a heart attack. In more than 20% of cases, the patient will progress to heart failure within 5 years [3,4,5,6]. Following the ischemic insult, a series of molecular and cellular physiological pathways are triggered to repair the damaged myocardium. As cardiomyocytes are post-mitotic cells, repair involves the replacement of necrotic myocytes with scar tissue [7]. When ischemia is prolonged and a significant portion of the myocardium is affected, MI may lead to alteration of left ventricle (LV) size, shape and function, a process termed LV remodeling [1,8]. Adverse LV remodeling constitutes the basis for ischemic heart failure [1].

A major determinant of adverse LV remodeling is the efficacy of infarct healing [9,10,11]. Current state-of-the-art treatment includes timely reperfusion of the coronary artery, along with angiotensin converting enzyme inhibitors, beta adrenergic receptor blockers, and statins. When effective, treatment limits infarct scar size and improves survival [9,12]. However, not all patients receive timely reperfusion therapy and not all patients respond to treatment with actual reflow in the artery, both of which significantly increase their risk of developing heart failure [9,12,13]. Improving the cardiac repair process in a predictable way is a strategy that may help to improve outcomes for this cohort [1,8].

MI wound healing involves three distinct phases that overlap: the inflammatory, proliferation, and maturation phases. During the inflammatory phase, recruited neutrophils and macrophages are pro-inflammatory (N1, M1), actively secreting proteases to clear necrotic cells and debris that are later replaced by a fibrotic scar. During the proliferation phase, neutrophils and macrophages convert from pro-inflammatory to reparative cell types (N2, M2) and support inflammation resolution, while myofibroblasts and endothelial cells are activated to form and revascularize the scar [1,8]. In addition to leukocytes, fibroblasts also exhibit a wide range of phenotypes, going from pro-inflammatory (F1) phenotype during the inflammation phase to anti-inflammatory (F2) phenotype during the proliferation phase [14]. During the maturation phase, the scar is further strengthened by cross-linking of the extracellular matrix (ECM) components, particularly collagen [1,8].

For optimal wound healing, balance among the different phases is crucial [1,15,16]. For example, a prolonged inflammatory phase may prevent or interfere with the proliferation and maturation phases, yielding an infarct too weak to support the structure of the LV and lead to LV aneurysm. Likewise, extending a robust reparative phase past the time of sufficient scar formation may increase total ECM deposition and yield an overly stiff LV, which could serve as a conduit for arrhythmias or promote diastolic dysfunction and lead to the development of heart failure [1,17].

ECM constituents play major roles in all phases of MI cardiac repair. Matrix metalloproteinases (MMPs) are a family of 25 proteolytic enzymes that collectively degrade all ECM components [18,19,20]. MMPs are major regulators of the ECM, conveyed into the infarct primarily by neutrophils and macrophages that begin infiltrating within minutes of the ischemic insult. In addition to leukocytes, MMPs are also produced by cardiomyocytes, fibroblasts and endothelial cells [19,20,21]. Of the MMPs studied in the myocardium, MMP-9 has received the majority of attention. Initially, this was due to the technical reason that MMP-2 and MMP-9 are the two MMPs visualized by gelatin zymography, and prior to the commercial availability of MMP antibodies, these two MMPs were the easiest to evaluate and, therefore, the most frequently studied. MMP-9 continues to be evaluated due to its high mechanistic connection with cardiac remodeling [20,22].

MMP-9 plasma levels correlate with MI mortality, LV remodeling and dysfunction across a variety of species and in humans [23,24,25,26]. Zhu and colleagues showed that higher plasma levels of MMP-9 predict in-hospital mortality in patients with acute MI, even after adjustment for all other risk factors [23]. Somuncu and colleagues showed that patients with MI who had MMP-9 plasma levels above 12.92 ng/mL at the time of hospital admission had 3.5-fold higher odds for cardiovascular mortality and increased risk for advanced heart failure compared to the group with lower MMP-9 concentrations [24]. High MMP-9 plasma levels during the first few hours of MI are associated with a lower ejection fraction and higher LV end-diastolic volume at discharge [26]. Similarly, higher levels of MMP-9 are reported in ruptured human left ventricles compared to control infarcts [27]. For this reason, we focus this review article on the role of MMP-9 in MI wound healing.

## 2. Regulation of MMP-9 Activity

In most cells, except neutrophils, MMP-9 is regulated at the transcriptional level by cytokines and growth factors (interleukin (IL)-13, tumor necrosis factor alpha (TNFα), transforming growth factor beta (TGFβ), vascular endothelial growth factor (VEGF)), and epigenetic mechanisms (histone modification, DNA methylation and non-coding RNA) [28]. MMP-9 in neutrophils is primarily regulated at the post-translational level, since preformed MMP-9 is stored in gelatinase granules and released upon neutrophil activation by inflammatory signals [19,29,30]. MMP-9 transcription is mediated through various transcription factors including nuclear factor κB (NFκB), transcription factor Sp1 (SP1) and activator protein 1 (AP1), which are highly responsive to inflammatory stimuli [22,31].

Transcribed MMP-9 is secreted in an inactive pro-form, consisting of an NH2-terminal pro-domain, a conserved catalytic domain, a linker domain, and a COOH-terminal hemopexin-like domain [32,33]. The catalytic domain contains a zinc ion that is essential for proteolytic activity and is highly conserved within the MMP family [28,34]. Interaction between the zinc ion and a cysteine residue on the NH2-terminal pro-domain masks the catalytic cleft, keeping the MMP-9 inactive [33,35,36]. Therefore, MMP-9 activation requires removal of the pro-domain or disruption of the zinc-cysteine interaction, known as the cysteine switch mechanism [28,33]. The most common route of MMP-9 activation is proteolysis of the pro-domain by other proteases such as MMPs -1, -2, -3, -7, or -13, cathepsin and plasmin [33,34,37,38,39,40,41,42,43]. MMP-9 can also be activated by post-translational modification of the pro-domain cysteine residue, including S-glutathionylation or S-nitrosylation [34].

There are a number of mechanisms built in to prevent excess or undesirable MMP-9 activation. In the circulation, alpha 2-macroglobulin prevents systemic activation of MMP-9 [44,45]. This abundant glycoprotein contains a unique amino acid sequence, which functions as a sequestering substrate for a broad range of active proteases. Upon MMP-9 proteolysis, alpha-2 macroglobulin undergoes conformational change to trap MMP-9 and mask its active site [45,46]. In addition, the conformational change exposes receptor binding domains that enable alpha 2-macroglobulin/MMP-9 complex binding to low density lipoprotein receptor related protein 1 (LPR1), resulting in clearance of MMP-9 from the circulation [45,47,48].

In tissue, tissue inhibitor of metalloproteases (TIMPs), a family of four proteins (TIMP-1, TIMP-2, TIMP-3, and TIMP-4), inhibit MMP-9 by forming a tight non-covalent complex with the catalytic site of the protease, thereby blocking substrate access [44,49,50]. TIMP-1 and TIMP-2 deficiency is associated with accelerated LV remodeling as a function of age as well as MI [51,52,53,54]. Fibroblasts and cardiomyocytes are the main source of TIMPs after MI [55]. In addition, most cells secrete variable amounts of TIMP-1 along with MMP-9 [56].

Active MMP-9 enzymatically cleaves numerous ECM substrates, including collagen, fibronectin and laminin, to facilitate ECM turnover and scar formation during cardiac wound healing [17,55]. Proteolysis of ECM substrates generates biologically active fragments, termed matricryptins, which regulate MI cardiac remodeling [44]. A list of MMP-9 generated matricryptins relevant to cardiac wound repair can be found in Table 1. In addition to ECM constituents, MMP-9 processes numerous cytokines and chemokines including TNFα, IL-1β, TGFβ, and CXC motif ligands (CXCL-1,4,5,7, and 12) [44,57,58,59,60,61]. Thus, MMP-9 is capable of propagating MI inflammatory signaling that is both necessary and potentially deleterious [62,63,64].

## 3. MMP-9 Signaling in the Inflammatory Phase

The inflammatory phase occurs primarily over the first week in the mouse model of permanent occlusion, with a slightly longer time-frame for humans [70,71,72]. This phase is characterized by robust increase in pro-inflammatory cytokine release and degradation of the ECM following myocyte necrosis [22]. Damage associated molecular patterns (DAMPs), such as high mobility group box-1, S100A8/9, fibrinogen, fibronectin, heat shock proteins, hyaluronic acids, ATP, complement, and RNA/DNA secreted from necrotic/injured cells and the damaged ECM, attract leukocytes to the infarcted LV. Activated leukocytes further release DAMPs to amplify the inflammatory response [1,73]. Neutrophils and monocytes are the predominant infiltrating cells after MI (Figure 1). They regulate tissue reprogramming by releasing various ECM degrading MMPs, serine proteases, chemokines, and cytokines [74,75,76,77].

MMP-9 is an early and major MMP brought into the infarct region by activated leukocytes within minutes of the ischemic insult [8,86]. Increased MMP-9 in the infarcted region has been demonstrated in various animal models including rat, mouse, rabbit, dog, and pig [22,64,87,88,89]. In mice, the highest increase is observed from day 1 to day 4, and corresponds to neutrophil and macrophage infiltration into the infarct [27]. Infiltrating neutrophils secrete preformed MMP-9, which initiates early removal of necrotic debris. Further, MMP-9 activates IL-1β, IL-8, and CXCL6 by proteolytic processing. These molecules form a significant positive feedback loop for neutrophil activation and chemotaxis resulting in sustenance of inflammation. MMP-9 can also regulate IL-8 activity by negative feedback as C-terminal cleavage causes IL-8 inactivation [90]. MMP-9 further assists in prolonging inflammation by cleaving CD36, which leads to inhibition of neutrophil apoptosis by lowering caspase-9 expression [91].

Along with neutrophils, early pro-inflammatory (M1) macrophages produce large amounts of MMP-9 after MI. MMP-9 overexpression specific to macrophages unexpectedly improves ejection fraction and blunts the inflammatory response in a mouse model of MI [92]. One possible mechanism through which MMP-9 may blunt inflammation is by cleaving the receptor for advanced glycation end products (RAGE) into soluble RAGE [93]. Soluble RAGE has anti-inflammatory properties and low levels of soluble RAGE in patients are associated with heart failure [94].

The role of B- and T-cells in post-MI wound healing is still unclear. In rat models of MI, CD8^+^ T-cells are activated and their cytotoxic actions have been demonstrated in vitro on healthy cardiomyocytes [1,95]. Ilatovskaya and colleagues recently showed that mice deficient in functional CD8^+^ T-cells had improved survival and cardiac physiology at day 7 after MI. However, same mice also had exacerbated inflammation, elevated MMP-9 levels and poor scar formation, which resulted in later cardiac rupture in 100% of CD8^+^ T-cell deficient mice compared to 33% of wild type mice [96]. MMP-9, by regulating calcium influx, coordinates CD4^+^ and CD8^+^ T-cell proliferation, and MMP-9 deletion reduces IL-2, TNFα, and interferon gamma (IFNγ) gene expression in these cells [31]. Similarly, B-cells are responsible for the inflammatory response by mobilization of pro-inflammatory monocytes after MI. B-cell depletion using CD20 antibodies reduced apoptotic cell numbers and prevented adverse cardiac remodeling [1,97].

There is a significant association between elevated MMP-9 in the infarct and intensified inflammatory cell infiltration in animal models of MI. However, various anti-inflammatory strategies initiated early in MI to limit neutrophil influx worsened cardiac physiology, despite reducing inflammation and acute injury [98]. MMP-9 deficiency yielded a reduction in LV rupture rates and leukocyte influx [87,90,99].

Cardiomyocytes and fibroblasts localized to the infarct region also secrete MMP-9 during the inflammatory phase [89]. Various conditions, such as hypoxia and aldosterone, elevate MMP-9 expression in cardiomyocytes, whereas peroxisome-proliferator-activated receptor β/δ (PPARβ/δ) activation decreases its expression by reducing reactive oxygen species production [100,101]. Pro-inflammatory (F1) fibroblasts can increase MMP-9 expression in response to ischemia and stress, though they are not the primary contributors. Increased MMP-9 in fibroblasts decreases collagen synthesis resulting in a net collagenolytic environment [14,44]. This is an important and necessary function, since a major role of the inflammatory phase is to degrade and clear the necrotic tissue to make room for a scar.

## 4. MMP-9 Signaling in the Proliferation Phase

The proliferation phase overlaps with later stages of the inflammatory phase, concluding about 2 weeks after MI in the mouse model [70]. This phase is characterized by formation of granulation tissue and consists of macrophages, myofibroblasts, new blood vessels, and ECM [14,102]. Transition from inflammatory to proliferation phase is dependent on the cardiac microenviroment, and leukocytes aid in inflammation resolution [1,103,104]. Phagocytosis of apoptotic neutrophils stimulates polarization of inflammatory macrophages to anti-inflammatory/reparative macrophages. Ingestion of apoptotic neutrophils reduces expression of pro-inflammatory cytokines, such as IL-1β and TNFα, while increasing expression of anti-inflammatory and pro-fibrotic cytokines, such as IL-10 and TGFβ in macrophages [105,106,107]. Furthermore, neutrophil gelatinase-associated lipocalin released from neutrophils also stimulates the polarization of inflammatory to reparative macrophages. Late stage neutrophils release annexin A1 and lactoferrin, which inhibit neutrophil migration and recruitment, as well as induce apoptosis of neutrophils. Apoptotic neutrophils express scavenger receptors, which bind and deplete inflammatory mediators, aiding inflammation resolution [1].

MMP-9 plays a dual role in inflammation resolution (Figure 2). MMP-9 stimulates inflammation by inhibiting neutrophil apoptosis and macrophage phagocytosis through the CD36 receptor [19]. CD36 is a class B scavenging receptor degraded by MMP-9. CD36 is also a marker of mature cardiomyocytes [108]. Intact CD36 stimulates neutrophil apoptosis and macrophage phagocytosis [91]. MMP-9 cleaves platelet glycoprotein 4 into several fragments that inhibit neutrophil apoptosis and macrophage phagocytosis [20]. MMP-9 is an M1 macrophage marker and by processing inflammatory molecules such as CXCL4, CXCL12, IL-8 and TGFβ1, it aids in anti-inflammatory/reparative M2 macrophage polarization [31]. Direct stimulation of macrophages with MMP-9 produces a mixed transition state of the M1–M2 phenotype with higher expression of CCL5 and lower expression of CCL3, IL-1β, IL-6 and TGFβ [20,109].

The hallmark of the proliferation phase is activation of fibroblasts that includes a temporally linear increase in the expression of α-smooth muscle actin [110]. Activated fibroblasts produce collagen necessary for mechanical support of the newly formed scar. Reparative macrophages secrete factors that are crucial for fibroblast activation and proliferation [110]. Many cells, including reparative macrophages produce and secrete latent TGFβ, which, in its active form, is a very potent suppressor of inflammation [31]. MMP-9 cleaves latent TGFβ into its active form, which in turn stimulates fibroblast migration and activation [14,19,111]. MMP-9 is also directly involved in promoting fibroblast migration. Treatment of cardiac fibroblasts with MMP-9 stimulates migration, increases collagen synthesis, and upregulates angiogenic factors [44,112]. Furthermore, MMP-9 generates ECM fragments that induce fibroblast migration and collagen synthesis. MMP-9 cleaves osteopontin at three different amino acid positions to generate four peptides. In vitro, two of these peptides increase fibroblast migration [20,69]. MMP-9 also releases insulin-like growth factor 1 (IGF-1) from its binding protein to induce collagen and integrin expression [111].

During the proliferation phase, revascularization of the newly formed scar occurs, which is important for oxygen and nutrient delivery. Angiogenesis is a complex process that depends on endothelial cells, smooth muscle cells, and their interactions with ECM and angiogenic factors [19]. MMP-9 can both promote and inhibit angiogenesis. Deletion of MMP-9 stimulates neovascularization, and MMP-9 inhibits apoptosis of endothelial cells in a chronic heart failure model [19,64,113]. Endostatin and tumstatin, released by MMP-9 processing of collagen XVIII and collagen IVα3, inhibit angiogenesis [65,66,67]. The C-1158/59 fragment, generated from collagen type Iα1 by MMP-9 and MMP-2, increases neovascularization in vivo. MMP-9 may further degrade C-1158/59 to inactivity, thus, inhibiting angiogenesis in a negative feedback loop. Indeed, the plasma level of MMP-9 is inversely related to the level of C-1158/59, both in humans and in mice [20,65,114]. MMP-9 processing of collagen IV may inhibit endothelial cell growth and migration as the cryptic regulatory peptide required for the process is exposed [115]. At the same time, MMP-9 favors endothelial cell proliferation and migration as it clears the surrounding ECM during the early stages of MI, and macrophage derived MMP-9 is involved in capillary branching [63,115,116]. Furthermore, MMP-9 is downstream of VEGF signaling that initiates CD34+ endothelial progenitor and stem cell migration and promotes angiogenesis during hypoxia [116].

## 5. MMP-9 Signaling in the Maturation Phase

The maturation phase takes place over the weeks to months after MI in the mouse model. During this phase, inflammation has receded in the infarct region and fibroblasts generate and maintain the newly formed scar [14]. The ECM is enzymatically cross-linked to further strengthen the scar and prevent rupture. Reparative cells are inactivated and undergo apoptosis or removal [1]. At this point, the number of fibroblasts in the infarct region is reduced compared to the proliferation phase. Fibroblasts remain indefinitely active, which is essential for the maintenance of homeostasis in the newly formed scar [14,117]. MMP-9 plays a role in the maturation of the infarct scar as well. In a mouse model of MI, MMP-9 deletion increased infarct scar stiffness by increasing lysyl oxidase activity and cross-linking, while paradoxically decreasing collagen deposition. Thus, MMP-9 deletion prevents LV dilation and rupture both by reducing collagen degradation and increasing cross-linking [118].

## 6. Future Perspectives and Conclusions

MMP-9 is involved in all phases of cardiac wound healing, is secreted by the majority of cell types present at the infarct, and has different effects on LV remodeling depending on timing and cell source. This explains why MMP-9 inhibition or overexpression in the experimental setting has generated seemingly inconsistent results. Global MMP-9 deletion in a mouse model of MI increased survival and improved cardiac repair and remodeling by attenuating LV dilatation, reducing macrophage infiltration, and limiting collagen accumulation [87,119]. At the same time, a selective MMP-9 inhibitor (MMP-9i), started 3 h after MI, showed detrimental remodeling associated with reduced ejection fraction, increased wall thinning and prolongation of the inflammatory response. The opposing results were attributed to the baseline differences between MMP-9 null and WT mice, and highlighted the importance of a translational approach when designing experiments [63]. Global MMP-9 null mice demonstrate a compensatory increase in MMP-3 and a decrease in MMP-14, which may explain some of the opposing results [64]. Elevated levels of MMP-14 are associated with increased MI mortality, while MMP-3 is associated with LV remodeling and heart failure [21].

MMP-9 regulates the activity of many cytokines, which in turn feed-back to influence the expression of MMP-9. One question that remains is how to differentiate between the actions of MMP-9 on substrates alone vs. MMP-9 effects to amplify the inflammatory response. While this problem is technically and conceptually challenging from a reductionist view and would require in vitro examination, it is also not translationally relevant because MI includes both signaling pathways that work in concert.

MMP-9 overexpression selectively in macrophages also improves cardiac repair by attenuating inflammation and increasing ejection fraction [92]. This tells us that MMP-9 may have a different role depending on the cellular source, in part due to various cell types secreting different substrates [20]. A greater focus on MMP-9 substrates may be a fruitful strategy in using MMP-9 as a target in cardiac repair. Using biologically active MMP-9 derived fragments, which have more specific roles in MI wound healing, is an alternative approach. In vivo administration of the collagen fragment C-1158/59 limited LV remodeling in mice by reducing LV dilation [65]. This strategy of targeting the downstream substrate, rather than MMP-9 itself, could be a useful means to accelerate resolution or stimulate cardiac repair. MMP-9 derived matricryptins during MI is a nascent research area ripe for investigation. MMP-9 has also been associated with respiratory failure in COVID-19 patients. While we know that plasma MMP-9 is increased in COVID-19 patients with severe respiratory syndrome, the cellular origin has yet to be assigned [120]. MMP-9 regulates T-cell and macrophage chemotaxis in viral myocarditis due to coxsackievirus B3 infection and may play a role in COVID-19 induced myocarditis [121,122]. The effect of COVID-19 on cardiac disease has been reviewed [123,124]. In summary, MMP-9 has a favorable or detrimental effect on cardiac wound repair depending on the cell source and timing of expression.

## Figures and Tables

**Figure 1 biomolecules-11-00491-f001:**
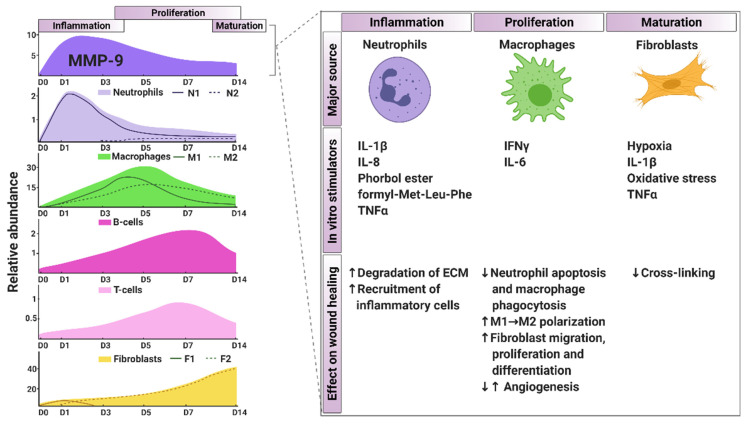
Temporal profile of MMP-9, leukocytes, and fibroblasts during myocardial infarction wound healing. The left side shows the temporal profile of MMP-9 and the relative abundance of neutrophils, macrophages, B-cells, T-cells, and fibroblasts during inflammation, proliferation and maturation phase. The graphs are based on the current literature [77,78,79,80,81,82,83,84]. The right side shows the cellular source of MMP-9, as well as in vitro stimulators of MMP-9 release and the effect of MMP-9 on different myocardial infarction wound healing processes. IL-1β: Interleukin 1 beta, IL-8: Interleukin 8, MMP-9: Matrix metalloproteinase-9, TNFα: Tumor necrosis alpha, N1: Pro-inflammatory neutrophils, N2: Anti-inflammatory/reparative neutrophils, M1: Pro-inflammatory macrophages, M2: Anti-inflammatory/reparative macrophages, F1: Pro-inflammatory fibroblasts, F2: Anti-inflammatory/reparative fibroblasts. Created with BioRender.com (accessed on 23 March 2021) [85].

**Figure 2 biomolecules-11-00491-f002:**
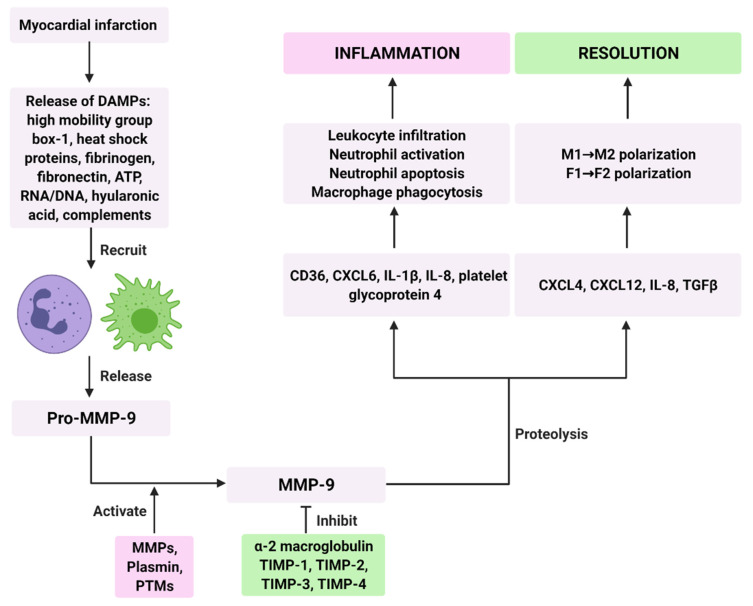
MMP-9 roles in inflammation and resolution after myocardial infarction. DAMPs: Danger associated molecular patterns, CD36: Cluster of differentiation 36, CXCL: CXC motif ligand, IL-1β: Interleukin 1 beta, IL-8: Interleukin 8, MMP: Matrix metalloproteinase, PTM: Post-translational modification, TIMP: Tissue inhibitor of metalloproteinases, TGFβ: Transforming growth factor beta. M1: Pro-inflammatory macrophages, M2: Anti-inflammatory/reparative macrophages, F1: Pro-inflammatory fibroblasts, F2: Anti-inflammatory/reparative fibroblasts. Created with BioRender.com (accessed on 23 March 2021) [85].

**Table 1 biomolecules-11-00491-t001:** Selection of myocardial infarction relevant matrix metalloproteinase 9 (MMP-9) derived matricryptins.

ECM Parent Protein	ECM-Fragment	Effect on Cardiac Wound Healing	Reference
Collagen I	C-1158/59	Stimulates neovascularization	[65]
Collagen IV	Tumstatin	Inhibits angiogenesis	[66]
Collagen XVIII	Endostatin	Inhibits angiogenesis	[67,68]
Osteopontin (OPN)	OPN-p151 OPN-p152	Increases fibroblast migration and wound healing	[69]

ECM: Extracellular matrix; OPN: osteopontin.

## Data Availability

Not applicable.

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
