# Peer review of "Infarct in the Heart: What’s MMP-9 Got to Do with It?"

_biomolecules, 2021, doi:10.3390/biom11040491_

Round 1

Reviewer 1 Report

The manuscript provides a concise yet comprehensive and insightful review of a major problem plaguing the field of MMPs and ADAMs biology: the fact that these proteinases exert positive as well as negative actions in the development of diseases, particularly, heart diseases. This duality of roles complicates the treatment of many relevant diseases, being a major reason behind the failure of preclinical and clinical trials targeting MMPs with inhibitors.  

The authors do a excellent job at presenting the multiple beneficial and detrimental roles played by MMP-9 in the settings of myocardial infarction mortality and left ventricular remodeling with a focus on cardiac wound healing. The authors correctly describe the need for efforts to precisely delineate the signaling events elicited by MMP-9-mediated proteolysis of substrates in the heart. Most importantly, the authors raise awareness of the importance of targeting MMP-9 only for its detrimental actions. 

An important contribution of this review to the field is the quite detailed description of the wound healing process (involving the inflammatory, proliferation and maturation phases). This is greatly appreciated as the role of MMP-9 in each of these phases has previously not been described in such a concise fashion.

The authors highlight very well the (often contradictory) findings between MMP-9 gene deletion and  MMP-9 activity blockade. This is a piece of information rarely recognized by investigators who are not in the field of MMPs. Conventional thinking advocates for gene deletion being equal to activity blockade. The experience of decades on research in the field of MMPs biology shows that, MMPs activity blockade rarely replicates MMPs gene deletion (and this is not just the case of MMP-9). Similarly, cell-specific MMP gene knockdown results in phenotypes that differ from global gene deletion or systemic activity blockade. 

I have only minor suggestions to improve this manuscript:

  1. This Special Issue was intent on addressing the roles of MMPs in heath and disease in the times of COVID-19. There is no obligation to cover the role of MMP-9 in the settings of cardiac disease in COVID-19 patients. Nonetheless, could the authors add either a couple of brief paragraphs on what has been published on the topic or at least couple of thoughts about what might be expected for MMP-9 contributions to cardiac disease (specifically, myocardial infarction) in COVID-19 patients which suffer from hyper-inflammation to which MMP-9 could contribute in the heart? 
  2. As correctly pointed out, MMPs (MMP-9 included) regulate the activity of many cytokines, which in turn influence the expression of MMPs. A question not fully answered is how to differentiate between the actions of MMP-9 on substrates with no impact on MMPs activity vs. inflammatory cytokines (e.g., TNF-a) whose expression can influence the expression of MMP-9?  Could the authors briefly express their view on how to target this technically and conceptually challenging problem in the setting of myocardial infarction?
  3. In their concluding statement, the authors suggest the potential of utilizing biologically active products of MMPs activity as condition-specific therapeutic tools. In the interpretation of this reviewer, proteolysis products, could present very elegant and brilliant solutions to the long-standing problem of how to target MMPs not only selectively but primarily 'only for their detrimental actions'.  For instance, if detrimental effects are caused by the production of a certain class of molecules; MMPs activity could be targeted using blockers or antagonists of these molecules (which would be an approach analogous to the one we normally undertake to assess other signaling pathways). Could the authors develop a bit more this concept?  Could the authors, perhaps, add a simple figure depicting how they foresee this approach would work for therapeutic purposes in the settings of myocardial infarction?

Author Response

Reviewer 1

  1. This Special Issue was intent on addressing the roles of MMPs in heath and disease in the times of COVID-19. There is no obligation to cover the role of MMP-9 in the settings of cardiac disease in COVID-19 patients. Nonetheless, could the authors add either a couple of brief paragraphs on what has been published on the topic or at least couple of thoughts about what might be expected for MMP-9 contributions to cardiac disease (specifically, myocardial infarction) in COVID-19 patients which suffer from hyper-inflammation to which MMP-9 could contribute in the heart? 

We have added to future perspectives on line 321: “MMP-9 has also been associated with respiratory failure in COVID-19 patients. While we know that plasma MMP-9 is increased in COVID-19 patients with severe respiratory syndrome, cellular origin has yet to be assigned.115 MMP-9 regulates T-cell and macrophage chemotaxis in viral myocarditis due to coxsackievirus B3 infection and may play a role in COVID-19 induced myocarditis.116, 117 The effect of COVID-19 on cardiac disease has been reviewed.122, 123

  1. As correctly pointed out, MMPs (MMP-9 included) regulate the activity of many cytokines, which in turn influence the expression of MMPs. A question not fully answered is how to differentiate between the actions of MMP-9 on substrates with no impact on MMPs activity vs. inflammatory cytokines (e.g., TNF-a) whose expression can influence the expression of MMP-9?  Could the authors briefly express their view on how to target this technically and conceptually challenging problem in the setting of myocardial infarction?

We have added on line 305: “MMP-9 regulates the activity of many cytokines, which in turn feed -back to influence the expression of MMP-9. One question that remains is how to differentiate between the actions of MMP-9 on substrates alone vs. MMP-9 effects to amplify the inflammatory response. While this problem is technically and conceptually challenging from a reductionist view and would require in vitro examination, it is also not translationally relevant because MI includes both signaling pathways that work in concert.”  

  1. In their concluding statement, the authors suggest the potential of utilizing biologically active products of MMPs activity as condition-specific therapeutic tools. In the interpretation of this reviewer, proteolysis products, could present very elegant and brilliant solutions to the long-standing problem of how to target MMPs not only selectively but primarily 'only for their detrimental actions'.  For instance, if detrimental effects are caused by the production of a certain class of molecules; MMPs activity could be targeted using blockers or antagonists of these molecules (which would be an approach analogous to the one we normally undertake to assess other signaling pathways). Could the authors develop a bit more this concept?  Could the authors, perhaps, add a simple figure depicting how they foresee this approach would work for therapeutic purposes in the settings of myocardial infarction?

We have added on line 317: “In vivo administration of the collagen fragment C-1158/59 limited LV remodeling in mice by reducing LV dilation.65 This strategy of targeting the downstream substrate, rather than MMP-9 itself, could be a useful means to accelerate resolution or stimulate cardiac repair.”  

Reviewer 2 Report

The manuscript entitled "Infarct in the heart: what’s MMP-9 got to do with it?" is an important state-of-the-art review, that shows the dual role of MMP-9 in post-infarctum myocardium, and suggests the best strategies to use this knowledge in the near future for the best benefit of patients.

I don't have any suggestions, since in my view the paper is excellent. In my view, the only change would be to make a separate paragraph when the authors describe the TIMPs, and maybe write a little bit more about their importance (line 119).

Good Work !

Author Response

Reviewer 2

  1. the only change would be to make a separate paragraph when the authors describe the TIMPs, and maybe write a little bit more about their importance (line 119).

We have added on line 122: “TIMP-1 and TIMP-2 deficiency is associated with accelerated LV remodeling as a function of age as well as MI.51-54

Reviewer 3 Report

In the current manuscript, the authors nicely point MMP-9 in the context of myocardial infarction, focusing on every aspect of the inflammatory, proliferative and maturation process, showing to the readers what is known about endogenous expression or genetic manipulation of MMP-9 in different animal models of MI. The authors aim to shed light about the role of MMP-9 but they achieved the opposite, although it is obviously not their fault. The main conclusion is that we still lack sufficient information about the precise role of MMP-9 towards therapeutical use, since in most cases “the mirror has two faces”, or at least results are species-specific, and how close or how different are some animal procedures compared to the pathogenesis of human myocardial infarction, should be discussed in order to draw solid conclusions.

Comments

  • Discussion about species-specific effect of MMP-9.
  • Contribution of the rest of MMP family members specially in those papers in which MMP-9 was deleted.
  • Check grammar.

Author Response

Reviewer 3

  1. Discussion about species-specific effect of MMP-9.

We have updated the manuscript to clarify which species is being discussed. We include information on MMP-9 in humans, mice, rats, pigs, rabbits and dogs after MI. 

  1. Contribution of the rest of MMP family members specially in those papers in which MMP-9 was deleted.

We have added on line 301: “Global MMP-9 null mice have a compensatory increase in MMP-3 and a decrease in MMP-14, which may explain some of the opposing results.62 Elevated levels of MMP-14 are associated with increased post-MI mortality while MMP-3 is associated with LV remodeling and heart failure.21

  1. Check grammar.

We have checked the grammar.